# Predictive Biomarkers of Immune Checkpoint Inhibitor Response in Breast Cancer: Looking beyond Tumoral PD-L1

**DOI:** 10.3390/biomedicines9121863

**Published:** 2021-12-08

**Authors:** Nan Chen, Nicole Higashiyama, Valentina Hoyos

**Affiliations:** 1Baylor College of Medicine, Houston, TX 77030, USA; nhigash@gmail.com (N.H.); valentina.hoyos@bcm.edu (V.H.); 2Lester and Sue Smith Breast Center, Baylor College of Medicine, Houston, TX 77030, USA; 3Department of Cell and Gene Therapy, Baylor College of Medicine, Houston, TX 77030, USA

**Keywords:** immunotherapy, triple-negative breast cancer, biomarkers

## Abstract

Immune checkpoint inhibitors utilize the immune system to kill cancer cells and are now widely applied across numerous malignancies. Pembrolizumab has two breast-specific indications in triple-negative disease. Currently, programmed death ligand-1 (PD-L1) expression on tumor and surrounding immune cells is the only validated predictive biomarker for immune checkpoint inhibitors (ICIs) in breast cancer; however, it can be imprecise. Additional biomarkers are needed to identify the patient population who will derive the most benefit from these therapies. The tumor immune microenvironment contains many biomarker candidates. In tumor cells, tumor mutational burden has emerged as a robust biomarker across malignancies in general, with higher burden cancers demonstrating improved response, but will need further refinement for less mutated cancers. Preliminary studies suggest that mutations in breast cancer gene 2 (*BRCA-2*) are associated with increased immune infiltration and response to ICI therapy. Other genomic alterations are also being investigated as potential predictive biomarkers. In immune cells, increased quantity of tumor-infiltrating lymphocytes and CD8+ cytotoxic T cells have correlated with response to immunotherapy treatment. The role of other immune cell phenotypes is being investigated. Peripherally, many liquid-based biomarker strategies such as PD-L1 expression on circulating tumor cells and peripheral immune cell quantification are being studied; however, these strategies require further standardization and refinement prior to large-scale testing. Ultimately, multiple biomarkers utilized together may be needed to best identify the appropriate patients for these treatments.

## 1. Immune Checkpoint Inhibition in Breast Cancer

Immune checkpoint inhibitors (ICIs) harness the ability of the immune system to recognize and destroy cancerous cells. They have demonstrated varied clinical success across a spectrum of solid organ malignancies. In comparison to tumors such as melanoma and lung, immunotherapy in breast cancer has only shown a modest benefit in certain subsets of patients. Initial studies with ICI monotherapy in advanced triple negative breast cancer (TNBC) showed disappointing results [1,2,3], necessitating combination therapy with other neoplastic agents [4,5]. Currently, the FDA has approved pembrolizumab in two indications for the treatment of breast cancer based on results of phase III clinical trials combining it with standard of care chemotherapy in a subset of patients with TNBC [4,6]. Pembrolizumab, a programmed cell death protein 1 (PD-1) inhibitor, was recently approved in combination with neoadjuvant chemotherapy for the treatment of high-risk early-stage TNBC [6] and first-line programmed cell death ligand 1 (PD-L1+) metastatic TNBC [6]. A similar indication was given to atezolizumab in combination with abraxane based on the results of the IMpassion 130 study [7]; however, the application for full approval was ultimately withdrawn. Additionally, pembrolizumab has two tumor agnostic indications which can be used for all unresectable solid organ malignancies, including breast cancer—high tumor mutational burden (TMB, defined as greater than 10 mutations per megabase) and deficient mismatch repair (dMMR) [8,9]. Given the heterogeneity of breast cancer, it is of great importance to appropriately identify the patients that will derive the most benefit from this class of drugs. Conversely, patients who would not benefit from ICIs can avoid their unique side effects, including long-term endocrinopathies and autoimmune diseases, which can ultimately impact patients’ overall health and quality of life.

Currently, the only predictive biomarker validated in phase III ICI trials in breast cancer is PD-L1 expression on tumor cells and surrounding cells as measured by immunohistochemistry (IHC). Pembrolizumab, like other ICIs, has an accompanying biomarker assay, 22C3, which looks at expression on tumor cells and neighboring immune cells and produces a combined score [10]. However, other ICIs utilize different accompanying assays, and there remain some disadvantages with the current usage of PD-L1 assays overall. Various studies have shown limited concordance between assays, limiting their clinical utility [11,12]. PD-L1 measurements from the IMpassion130 trial were obtained using the SP142 assay and compared with measurements using other commercially available assays. Ultimately, the results from the various assays could not be harmonized, suggesting different assays are identifying unique populations of patients [12]. Because of these inconsistencies, it is difficult to compare one assay to another. In practice, multiple assays per patient may be sent, which increases tissue requirements and the overall cost of care. In addition to the standardization of PD-L1 evaluation in tissue, other biomarkers are needed to strengthen our ability to determine immunotherapy response. As the number of FDA-approved ICIs increases and additional drugs targeting other molecules in the checkpoint pathway move further in clinical development, it is imperative for clinicians to be able to use ICIs judiciously. This review focuses on the current landscape of both tumor-based and liquid-based potential biomarkers for ICI therapy.

## 2. Tumor Cell Microenvironment Biomarkers

### 2.1. Tumor Mutational Burden (TMB)

The tumor microenvironment (TME) is a dynamic milieu of tumor cells, immune cells, and stromal cells which can be altered by a host of changes, including tumoral mutations and systemic cancer therapies. Within this environment, high TMB tumors can generate a greater variety of neoantigens, leading to a more robust immune response [13]. Due to this, it has been evaluated as a potential predictive and prognostic biomarker for immunotherapy. Subgroup analysis from the KEYNOTE-158 trial in which patients were treated with pembrolizumab monotherapy was associated with a robust 29% overall response rate in high TMB patients (≥10 mutations per megabase), although this trial did not include breast cancer patients [9]. Biomarker analysis of the KEYNOTE-086 study comprised of metastatic TNBC patients treated with pembrolizumab monotherapy did show a positive association between high TMB (≥175 mutations per exome) and overall response rate, progression-free survival, and overall survival [14]. Furthermore, single-center data from the group at Dana Farber showed a significant difference in progression-free survival in patients with high TMB (≥10 mutations per megabase) within their metastatic TNBC cohort [15]. They also found that patients with high TMB were over three times more likely to respond to ICI [15]. Furthermore, they successfully showed that these tumors had increased neoantigen load and T cell infiltration, which are also associated with improved ICI response [16]. In a phase Ib study treating metastatic breast cancer with durvalumab, an anti-PD-L1 and tremelimumab, an anti-cytotoxic T lymphocyte-associated protein 4 (CTLA-4) therapy, responders also had a significantly higher mutational burden than non-responders, although they were not defined as high TMB [17]. At Memorial Sloan Kettering, they performed a large retrospective analysis on all of their patients who received either single-agent or combination immunotherapy at their center. Within the subset of breast cancer, they isolated the patients with the top 10% of TMB and demonstrated improved overall survival in this group as compared to the other 90% with a trend towards significance [18]. This trend was interestingly most notable in the ER+ subtype [18]. In contrast, the TAPUR trial included 28 breast cancer patients with high TMB (≥9 mutations per megabase) and showed an objective response rate of 21%, but subgroup analysis did not demonstrate that high TMB was a positive prognostic biomarker for treatment with ICIs [19]. Notably, the study was not powered to detect an association between TMB and progression-free survival (PFS). Additionally, a large dataset from The Cancer Genome Atlas (TCGA) showed that high TMB could not predict response to immunotherapy in certain cancers, including breast, which did not exhibit a corresponding increase in CD8+ T cell infiltration into the TME [20]. The study also found that TMB-High breast cancer patients did poorly with immunotherapy as compared to patients with lower TMB [20].

Overall, TMB is a robust predictive marker for immunotherapy response in metastatic breast cancer; however, it will need refinement to improve its validity. The definition of TMB across all cancers varies and may need to be adjusted for less mutated cancers such as breast.

### 2.2. DNA Repair Pathways

Mutations in DNA repair pathways, such as mismatch repair and homologous recombination, have also been investigated as predictive biomarkers for ICI due to the increased mutations and neoantigen production that are associated with impaired DNA repair [21]. DNA mismatch repair involves the correction of mispaired bases that occur during DNA replication. Errors commonly occur in areas of non-coding short-tandem repeats known as microsatellites, so dMMR due to mutations or epigenetic silencing of *MSH2, MLH1, MSH6, PMS2,* and *EPCAM* leads to microsatellite instability [22]. Deficient mismatch repair has been observed in multiple cancers and first demonstrated utility as a predictive biomarker for immunotherapy response in colorectal cancer [23]. After the success in colorectal cancer, KEYNOTE-158 demonstrated success in other solid organ malignancies with dMMR, including five cases of breast cancer [8]. However, as compared to other malignancies, breast cancer has a low overall prevalence of dMMR status, with studies citing a range of 0.5–1.5% [24]. Furthermore, a retrospective study conducted in TNBC did not find any association between dMMR status and PD-L1 expression or overall survival [25]. Nonetheless, dMMR breast cancers can be treated effectively with immunotherapy and should be considered for the appropriate patient and tumor biology [26].

Studies have also examined breast cancer gene 1 and 2 (*BRCA-1)* and (*BRCA-2)* mutations within the homologous recombination pathway as predictors of ICI response. Using large breast cancer datasets, researchers were able to show that, unsurprisingly, *BRCA-1* and *BRCA-2* mutated cancers did have an increased proportion of homologous recombination deficiency (HRD) genetic signatures and overall mutational load compared to BRCA proficient cancers [26]. Analysis on HRD deficient cancers overall, including breast, show a positive response to ICIs as compared to HRD proficient tumors [27]. However, despite both acting in the HRD pathway, *BRCA-1* and *BRCA-2* mutations have different effects on immunity. Increased PD-L1 expression on tumor cells was noted on *BRCA-1* mutated cancers only, and this difference may be related to an increased frequency of loss of *PTEN* function in *BRCA-1* as compared to *BRCA-2* [28]. In another study, upregulation of T cell, NK cell, and dendritic cell expression was noted in *BRCA-2* deficient tumors specifically [27]. Furthermore, *BRCA-2* mutated cancers demonstrated an improvement in overall survival when treated with ICIs in an MSKCC cohort [29]. While the role of *BRCA-1* and *2* mutations on the immune system have not been fully elucidated, preliminary studies show distinct differences which may ultimately impact clinical practice. Overall, the role of MMR and HRD mutations in breast cancer as predictive or prognostic biomarkers remains unclear and is an important area of further study.

## 3. Other Genomic Alterations

Various genomic alterations have also been investigated as candidate biomarkers. In other solid organ malignancies, mutations in important oncogenes such as *BRAF*, *STK11*, and *MDM2* have had an effect on the likelihood of response to immunotherapy; however, these occur very sparingly in breast cancer and with reported rates of less than 1% [30]. One study examined the genomic mutational landscape of over 3000 metastatic breast cancers and noted the most frequent mutations occurred in *PIK3CA*, *FGFR1*, and *PTEN;* however, there was no difference in mutational status between ICI responders and non-responders [30]. *PTEN* loss has been shown as an overall negative prognostic marker for breast cancer, both in overall survival and disease-free survival across breast cancer subtypes [31]. Additionally, it is present in more aggressive subtypes, such as HER2+ and triple-negative [31,32]. *PTEN* alterations have independently been examined further as a predictive biomarker of ICI response. *PTEN* knockdown can result in increased PD-L1 expression and decreased proliferation of T cells [32]. However, even with increased PD-L1 expression, patients with *PTEN* alterations can still have a poor response. Among a single-center cohort of patients with metastatic TNBC, patients with *PTEN* alterations had a significantly lower response rate to immunotherapy monotherapy or immunotherapy in combination with chemotherapy [15]. The small, phase II FUTURE-C-PLUS study also found that somatic mutations of *BACA1*, *KAT6A*, and *PKD1* were potential predictive biomarkers in metastatic TNBC patients treated with a combination of camrelizumab (anti-PD-1 antibody) and abraxane [33]. It is difficult to conduct formalized studies on these rare mutations in breast cancer, however in clinical practice, these genomic alterations, if present, may be used for individualized treatment plans incorporating ICI.

Additional ways to evaluate PD-L1 other than IHC have also been studied. The phase II SAFIR02 BREAST IMMUNO study looked at the efficacy of durvalumab compared to standard chemotherapy, and a correlative study examined how PD-L1 copy number alterations affected treatment [34]. The majority of patients did not have any copy number loss or gain, but the 20% of patients with copy number gain derived the most benefit from durvalumab monotherapy with a corresponding improvement in overall survival (OS) [34].

While the utility of individual genomic alterations as predictive biomarkers for ICIs has been limited by its frequency, more widespread usage of ICIs in breast cancer and larger analyses will hopefully provide additional data.

## 4. Immune Cell Microenvironment Biomarkers

Tumor-infiltrating lymphocytes (TILs) are lymphocytes identified on the tumor biopsy specimen. The prognostic value of TILs has already been demonstrated in the HER2+ and triple-negative subtypes [35], and there is continued exploration into its value as a predictive biomarker for immunotherapy. Using locally advanced breast cancer samples, retrospective analyses demonstrated a positive correlation between PD-L1 expression and TILs, although the mechanism of this interaction remains unknown [36]. Furthermore, higher levels of TILs in metastatic TNBC have been associated with improved response to pembrolizumab [37]. More specific immune components of the TME also serve as potential biomarkers.

ICIs work effectively as cytotoxic T lymphocytes become activated to recognize and destroy tumor cells. However, there are many other immune cells, some of which are immune-activating and others that are immune suppressive, that are involved in the ICI response. The presence of CD8+ cytotoxic T cells has been studied as a prognostic biomarker with increasing levels as seen on immunohistochemistry associated with improved survival in Estrogen Receptor negative (ER-) tumors [38]. Another study similarly noted that when CD8+ T cell gene expression was measured in the TCGA cohort, TNBC tumors with a higher presence of CD8+ T cells were associated with improved OS, and this effect was most prominent in patients with synchronous elevated expression of CD4+ and CD8+T cells [39]. In the phase II FUTURE-C-PLUS study, patients with immunomodulatory subtype (defined as the presence of CD8+ T cells on IHC) metastatic TNBC were selected for combination treatment, including camrelizumab, and demonstrated an impressive 81% overall response rate to ICI combination therapy [33]. Using samples from the IMpassion 130 study, increased CD8+ T cell localization in the immune inflamed subset demonstrated improved progression free survival (PFS) and OS after treatment with atezolizumab as compared to the excluded and desert subsets with less localization [40]. 

FoxP3+ CD4+ T regulatory lymphocytes play a vital role in regulating T cell differentiation and development via induction of T cell anergy and apoptosis, as well as via impairment of antigen-presenting cell (APC) activation through consumption and production of various cytokines [41]. ICIs are able to target T cell anergy by inhibiting the PD-1 and PD-L1 interaction. In many malignancies, increasing levels of FoxP3+ CD4+ T cells in the TME have been identified as a poor predictive and prognostic biomarker with lower levels of response and OS [41,42]. However, in breast cancer specifically, West et al. noted that in patients with ER- tumors, FoxP3+ CD4+ T cells were associated with improved recurrence-free survival, though this was only significant among high-grade tumors [43]. One potential mechanism of improved response may be related to concurrent increases in CD4+ regulatory T cells and CD8+ cytotoxic T cells since they noted a correlation between increasing amounts of CD4+ regulatory T cells and CD8+ cytotoxic T cells, and the improvement in response was dependent on this association [43]. 

Other studies have also evaluated the relationship between CD8+ cytotoxic T cells, CD4+ T cells, and immunotherapy response in breast cancer. In an evaluation of the patients in the I-SPY2 trials, a T/B-cell co-expression module looking at overall immune signatures was predictive of response to pembrolizumab [44]. This correlation did not hold true when more specific immune signatures targeted towards subsets of T cells were analyzed [44]. In an analysis of the KEYNOTE-086 patients, RNA signatures of an 18-gene inflamed T cell phenotype were positively correlated with response to pembrolizumab [45]. With these patients, responders also had increased staining of CD8+ T cells [14]. Additional gene panels reflecting increased immune activity have also been studied. The Functional Hotness Score incorporates gene expression levels of CD8+ cytotoxic T cells, granzyme B, and various chemokines which attract T cells into the TME, and higher scores were shown to be a positive prognostic biomarker for patient survival [46]. 

As neoantigen load and TMB increase in tumor cells, it increases the likelihood that one of the abnormal epitopes will be recognized by a T cell as a foreign antigen. Increased T cell receptor (TCR) diversity could also increase the likelihood of igniting an immune response. Low TCR diversity combined with lymphopenia at treatment initiation was associated with poorer overall survival [47]. A phase Ib study investigating the combination of durvalumab and tremelimumab in metastatic breast cancer collected data on TCR sequences prior to treatment and after two months of therapy [17]. They found that the total number of T cell clonotypes was increased in responders compared to non-responders [17]. They additionally noted that responders had increased expression levels of CD8 and granzymes [17]. Another small study treating patients with early-stage breast cancer with a combination of cryoablation and ipilimumab examined both T cell density and T cell clonality [48]. In patients who received the combination treatment, they noted an increased influx of T cells into the tumor microenvironment as compared to patients who received cryoablation alone. Furthermore, they showed that these patients had a smaller number but a larger diversity of T cell clones [48]. These findings support the usage of TCR clonality and expansion as a possible predictive marker of response to immunotherapy.

Myeloid cells in the TME, such as myeloid-derived suppressive cells (MDSCs) and tumor-associated macrophages, have been investigated as predictive biomarkers primarily in other malignancies. In melanoma, higher levels of MDSCs were associated with a poorer response in patients who were refractory to ipilimumab and subsequently treated with nivolumab [49]. Tumor-associated macrophages have been identified in many breast cancer biopsy samples, and while higher levels corresponded with more aggressive clinicopathologic features such as higher tumor grade and ER/PR negativity, it was ultimately not found to be an independent prognostic factor across all subtypes [50]. 

MHC (major histocompatibility complex) II expression has been examined among the patients in the I-SPY2 trial with pembrolizumab. They noted that across all patients treated with pembrolizumab, there was a positive correlation with all HLA class II molecules and response to treatment, an association which was not seen in the control arm of the study [51]. 

The immune cells of the TME represent a rich field of potential biomarkers. In order to apply its usage more broadly, we need a more nuanced understanding of the interaction between immune cell subtypes with cancer and immune cell subtypes with each other. While currently premature, this remains an area of significant research interest and effort.

## 5. Peripheral Blood Biomarkers

As a source of biomarkers, fresh or paraffin-embedded tissues have limitations. They are obtained invasively and thus are unable to be obtained frequently. Additionally, biopsy samples are only small representations of either larger tumors or multiple tumor sites and may not reflect overall tumoral heterogeneity. PD-L1 expression on tumors may differ between sites of metastatic disease [52]. Analysis of whole blood samples is easier to obtain, can provide information on multiple metastatic sites, and provide a more nuanced view of a patient’s overall cancer burden. Furthermore, studies examining PD-L1 expression in peripheral blood and matched tumor cells demonstrated poor concordance and measurement of both can provide additional predictive and prognostic information [53].

### 5.1. Circulating Tumor Biomarkers

PD-L1 expression in peripheral blood has been an area of targeted interest. Multiple methods have been developed to accurately measure PD-L1 expression on circulating tumor cells (CTCs), including the FDA-approved Cellsearch system [54] and reverse transcription droplet digital polymerase chain reaction (RT-ddPCR) [55], which allows for the measurement of very small amounts of RNA transcripts. In patients with detectable CTCs, both methods were able to reliably detect PD-L1 in a majority of patients of the small samples tested [54,55]. While PD-L1 in tumor cells is a well-established positive marker, in the peripheral blood, this does not hold true. A Chinese study noted that high PD-1 mRNA levels, as measured in peripheral immune cells of metastatic breast cancer patients, was a poor prognostic factor for PFS [56]. PD-L1 mRNA has also been measured from extracellular exosomes via ddPCR [57]. TNBC patients were treated with atezolizumab and abraxane, and peripheral blood was obtained at baseline and two months into treatment. In responders, they were able to note a decrease in exosomal PD-L1, and this was associated with a significant difference in PFS [57]. Furthermore, they noted an increased amount of natural killer (NK) cells in responders at baseline and a subsequent increase after treatment, implicating another cytotoxic immune cell that may hold promise as a predictive biomarker [57]. Another group utilized RT-PCR to analyze pretreatment levels of PD-1 mRNA in peripheral blood in patients with early-stage breast cancer and found that it correlated negatively with surgical outcomes [58] and ultimately disease-free survival [59]. 

Circulating tumor DNA (ctDNA) may also play a role in the evaluation of immunotherapy response. Patients with solid organ malignancies (including TNBC) treated with pembrolizumab had ctDNA measurements at baseline and while on therapy [60]. Across all tumor types, lower levels of ctDNA at cycle 3 of treatment compared to samples collected prior to therapy were associated with clinical response [60]. In breast cancer, the I-SPY2 trial analyzed ctDNA in patients treated neoadjuvantly with pembrolizumab-containing regimens and noted that early clearance of ctDNA was positively associated with pathologic complete response at the time of surgery [61]. While limited data is available, increased availability and accuracy of ctDNA testing make it an attractive potential biomarker.

Lactate dehydrogenase (LDH), a laboratory marker of cellular turnover, has also been analyzed as a potential predictive biomarker in breast cancer immunotherapy. Similar to other solid organ malignancies, elevated LDH has been recognized as a poor prognostic marker [62]. In the KEYNOTE-012 study, patients with greater than two-fold elevations in LDH had a poorer response to pembrolizumab monotherapy as compared to patients with more normalized LDH values [3]. A similar trend was noted in the KEYNOTE-086 study, with an improved overall response rate (ORR) in patients with normal LDH as compared to patients with elevated LDH [63]. This difference in ORR was not statistically significant.

Lastly, microRNA (miRNA) are single-strand non-coding portions of RNA that have a role in the post-transcriptional regulation of gene expression [64]. miRNA, due to their size, do not get degraded by RNase and are thus stable in plasma. Numerous miRNAs have been found to be upregulated in the peripheral blood of patients with TNBC; however, no studies, to our knowledge, have analyzed them as a predictive biomarker for immunotherapy in breast cancer [64]. Nonetheless, they remain exciting as potential biomarkers to be investigated in the future.

### 5.2. Circulating Immune Cells Biomarkers

Groups have also been interested in studying the changes in various types of immune cells, especially T cells, in the peripheral blood as potential biomarkers. Liu et al. treated advanced TNBC patients with a combination of camrelizumab (non-FDA approved PD-1 inhibitor) and apatinib (anti-vascular endothelial growth factor 2 (VEGF2) small molecule inhibitor) and looked at immune dynamics in pretreatment and post-treatment samples [65]. Responders had a higher level of CD4+ T cells at baseline, a higher increase of CD8+ T cells in post-treatment samples, and a higher level of B cells as compared to patients with stable disease or progression [65]. However, baseline proportions in these three cell groups were not found to be significant as predictive or prognostic biomarkers [65]. Efforts have also been made to determine if the presence and amount of specific markers on T cells in addition to their phenotypic markers could be used as biomarkers. In patients treated with combination tremelimumab and exemestane, a specific subset of CD4+ T cells demonstrated increased expression of inducible co-stimulator (ICOS), which is a marker of activated T cells and present at very low levels in naïve ones [66]. Interestingly, this observation did not hold true for CD8+ T cells [66]. This data implies that we may more accurately utilize immune cell phenotypes as predictive biomarkers with additional information about the activity rather than the presence of the cell.

PD-L1 expression can also be analyzed in peripheral immune cells. Analyzing PD-L1 expression on monocytes, researchers were able to correlate increased expression with increased cancer stage and tumor burden [67]. In a small study in metastatic ER+ breast cancer patients receiving a combination of tamoxifen, vorinostat, and pembrolizumab, increased PD-L1 expression on T cells (exhausted phenotype) was associated with improved progression-free survival with a trend towards improved overall survival [68].

## 6. Gut Microbiome

The gut microbiome is a novel area of investigation for oncologists. Data suggesting that increased microbiome diversity, most notably in colorectal cancer, can affect individual patient response to immunotherapy has opened up a new avenue of exploration [69]. There are multiple ongoing studies investigating the gut microbiome as a therapeutic target in breast cancer patients. NCT03358511 is investigating how the administration of probiotics affects CD8+ T cells in locally advanced, post-menopausal breast cancer patients [70]. While certain species of bacteria have been identified in the gut microbiome of lung and kidney cancer patients as associated with resistance to ICIs, these studies have thus far not been conducted in breast cancer [71]. In the future, the composition and the diversity of the gut microbiome may be incorporated as a predictive biomarker, but its current use remains premature.

## 7. Conclusions

Immunotherapy is a growing portion of the breast oncologist’s armamentarium. While the responses have been modest compared to other more immunogenic malignancies, it remains an attractive treatment option, especially for disease subtypes that were previously very difficult to treat, such as TNBC. It is urgently important to develop means to more accurately determine which patients will derive benefit from this therapy. Especially as these drugs move into the neoadjuvant arena, defining the right populations is critical since long-term immune-related adverse events need to be considered when treating patients in the curative setting. Currently, PD-L1 expression on tumor samples remains the only FDA-approved predictive biomarker. For breast cancer, we will need a more robust methodology incorporating more tumor-based and even blood-based biomarkers to fully understand the immunogenicity and vulnerabilities in each individual patient to immune checkpoint blockade. Ultimately, our surrogate biomarkers will need to provide an accurate and reproducible idea of the immune microenvironment. An algorithm incorporating multiple biomarkers, including PD-L1, may be better than our current clinical practice [72]. This review summarizes the currently studied options, which span from tumor-based biomarkers to liquid-based biomarkers to the gut microbiome (Figure 1). While many of these technologies are not fully developed and validated, we are hopeful that they can make their way into everyday clinical practice.

## Figures and Tables

**Figure 1 biomedicines-09-01863-f001:**
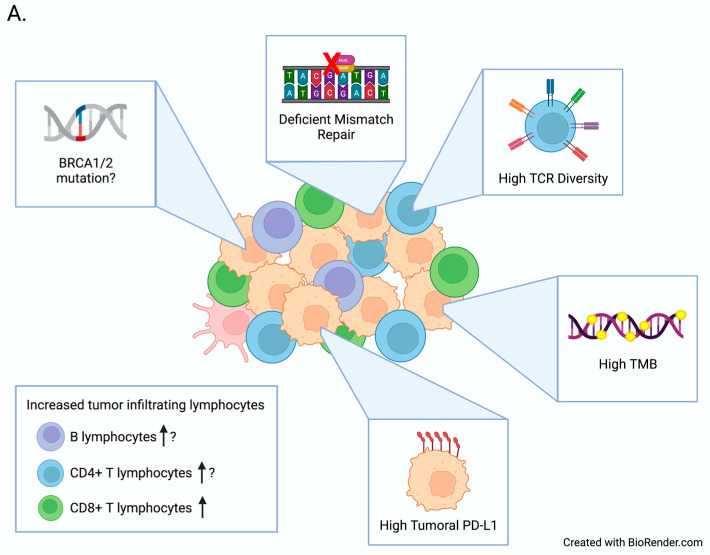
(**A**) Biomarkers of positive response to immune checkpoint inhibitors in the tumor microenvironment. (**B**) Biomarkers of poor response to immune checkpoint inhibitors in the tumor. Abbreviations: tumor mutational burden (TMB) programmed death ligand 1 (PD-L1) T cell receptor (TCR) breast cancer gene (BRCA).

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
