# Peer review of "Predictive Biomarkers of Immune Checkpoint Inhibitor Response in Breast Cancer: Looking beyond Tumoral PD-L1"

_biomedicines, 2021, doi:10.3390/biomedicines9121863_

Round 1
Reviewer 1 Report
In this manuscript, the authors have provided a collective information on the Biomarkers of immune checkpoint inhibitors in breast cancer. This is an interesting topic. But, the following points should be addressed before reconsidering for publication,
- The title is similar to a recently published original manuscript by Sivapiragasam et al., Cancer Med. 2021 doi: 10.1002/cam4.3550. Epub 2020 Dec 12. The authors should change the title.
- Separate sections on the following will improve the manuscript, a. Immune checkpoint inhibition b. Biomarkers of circulating tumor DNA c. Immunotherapy in breast cancer d. More information on gut and breast microbiome.
- A couple of figures should be included to summarize the biomarkers and targets.
Author Response
In this manuscript, the authors have provided a collective information on the Biomarkers of immune checkpoint inhibitors in breast cancer. This is an interesting topic. But, the following points should be addressed before reconsidering for publication,
- The title is similar to a recently published original manuscript by Sivapiragasam et al., Cancer Med. 2021 doi: 10.1002/cam4.3550. Epub 2020 Dec 12. The authors should change the title.
Thank you for bringing this manuscript to our attention. We have changed the title of our review to “Predictive Biomarkers of Immune Checkpoint Inhibition Response in Breast Cancer: Looking Beyond Tumoral PD-L1“.
- Separate sections on the following will improve the manuscript, a. Immune checkpoint inhibition b. Biomarkers of circulating tumor DNA c. Immunotherapy in breast cancer d. More information on gut and breast microbiome.
Thank you for this suggestion. The introduction has been renamed “Immune Checkpoint Inhibition in Breast Cancer” to better reflect the content of this section. Peripheral blood biomarkers has been further subdivided into Circulating Tumor Biomarkers and Circulating Immune Cells. Unfortunately, the gut microbiome remains an emerging area of research in breast immunotherapy and there is limited evidence for its use as a predictive biomarker for ICIs in breast cancer.
- A couple of figures should be included to summarize the biomarkers and targets.
Thank you for this suggestion. We have added an illustrated figure to our review that summarizes the positive and negative predictive biomarkers which have the most evidence.
Reviewer 2 Report
The current review is well written, highly informative and comprehensive.
I would recommend the authors to include and discuss in paragraph 5.2. the following peripheral blood (PB) parameters/analytes associated with immunotherapy outcome in breast cancer:
- Levels of serum lactate dehydrogenase (LDH) in blood (see KEYNOTE-012 and KEYNOTE-086 trials).
- ctDNA presence (I-SPY 2 trial) and ctDNA genomic analysis (please check in regards with genome instability number, mutations, or DNA methylation).
Also please update the same paragraph regarding the clinical value of PD-L1 assessment in the PB. I would recommend to include the following study (PMID: 32041353) which shows that PD-L1 expression on CTCs is associated with poor survival in metastatic breast cancer patients, and underlies a significant discordance in PD-L1 expression (on tumor cells and immune cells) between the PB and tumor tissue samples. These findings highlight the role of liquid biopsy in the assessment of PD-L1 and other potential biomarkers.
Author Response
I would recommend the authors to include and discuss in paragraph 5.2. the following peripheral blood (PB) parameters/analytes associated with immunotherapy outcome in breast cancer:
- Levels of serum lactate dehydrogenase (LDH) in blood (see KEYNOTE-012 and KEYNOTE-086 trials).
- ctDNA presence (I-SPY 2 trial) and ctDNA genomic analysis (please check in regards with genome instability number, mutations, or DNA methylation).
Thank you for both of these suggestions. We have added this section accordingly:
Circulating tumor DNA (ctDNA) may also play a role in the evaluation of immunotherapy response. Patients with solid organ malignancies (including TNBC) treated with pembrolizumab had ctDNA measurements at baseline and while on therapy.59 Across all tumor types, lower levels of ctDNA at cycle 3 of treatment compared to basement were associated with clinical response.59 In breast cancer, the I-SPY2 trial analyzed ctDNA in patients treated neoadjuvantly with pembrolizumab containing regimens and noted that early clearance of ctDNA was positively associated with pathCR at the time of surgery.60 While limited data is available, increased availability and accuracy of ctDNA testing makes it an attractive potential biomarker.
Lactate dehydrogenase (LDH), a laboratory marker of cellular turnover, has also been analyzed as a potential predictive biomarker in breast cancer immunotherapy. Similar to other solid organ malignancies, elevated LDH has been recognized as a poor prognostic marker.61 In the KEYNOTE-012 study, patients with greater than two fold elevations in LDH had a poorer response to pembrolizumab monotherapy as compared to patients with more normalized LDH values.3 A similar trend was noted in the KEYNOTE-086 study with an improved overall response rate (ORR) in patients with normal LDH as compared to patients with elevated LDH.62 This difference in ORR was not statistically significant.
Also please update the same paragraph regarding the clinical value of PD-L1 assessment in the PB. I would recommend to include the following study (PMID: 32041353) which shows that PD-L1 expression on CTCs is associated with poor survival in metastatic breast cancer patients, and underlies a significant discordance in PD-L1 expression (on tumor cells and immune cells) between the PB and tumor tissue samples. These findings highlight the role of liquid biopsy in the assessment of PD-L1 and other potential biomarkers.
Thank you for bringing this manuscript to our attention. We have added this sentence accordingly: “Furthermore, studies examining PD-L1 expression in peripheral blood and matched tumor cells demonstrated poor concordance and measurement of both can provide additional predictive and prognostic information.”
Round 2
Reviewer 1 Report
The manuscript was improved by revisions and can be accepted for publication.
Reviewer 2 Report
The manuscript is now more informative and scientifically sound.